# Factors Associated with Insured Children’s Use of Physician Visits, Dentist Visits, Hospital Care, and Prescribed Medications in the United States: An Application of Behavioral Model of Health-Services Use

**DOI:** 10.3390/ijerph21040427

**Published:** 2024-03-31

**Authors:** Tyrone C. Cheng, Celia C. Lo

**Affiliations:** 1Little Hall, School of Social Work, University of Alabama, Tuscaloosa, AL 35401, USA; 2Defense Personnel and Security Research Center, Peraton, Seaside, CA 93955, USA; clo@peraton.com

**Keywords:** children, health insurance, physician visits, dentist visits, hospital care, prescription use

## Abstract

This study is the first to examine factors in the utilization of physician services, dentist services, hospital care, and prescribed medications focusing exclusively on insured children in the United States. Data describing 48,660 insured children were extracted from the 2021 National Survey of Children’s Health. Children in the present sample were covered by private health insurance, public health insurance, or other health insurance. Logistic regression results showed self-reported health to be negatively associated with physician visits, hospital-care use, and prescription use, but teeth condition to be positively associated with dentist visits. Physician visits were associated negatively with age, Hispanic ethnicity, Asian ethnicity, family income at or below 200% of the federal poverty level, and other health insurance, but positively with parental education and metropolitan residency. Dentist visits were associated positively with girls, age, and parental education, but negatively with Asian ethnicity and public health insurance. Use of hospital care was associated negatively with age and Asian ethnicity, but positively with parental education and public health insurance. Use of prescriptions was associated positively with age, Black ethnicity, parental education, and public health insurance, but negatively with Hispanic ethnicity, Asian ethnicity, and family income at or below 200% of the federal poverty level. Implications included the expansion of public health insurance, promotion of awareness of medicine discount programs, and understanding of racial/ethnic minorities’ cultural beliefs in health and treatment.

## 1. Introduction

In the United States in 2022, 54.3% of children had private health insurance and 43.7% were covered by Medicaid; only 4.2% of children were uninsured [1]. However, being insured by any type of health insurance is not equivalent to children’s access to healthcare services [2]. Prior studies on the general population (including both insured and uninsured children) [3,4,5,6] have reported that 68.0–87.1% of American children had visited a physician and 36.3–79.5% had visited a dentist. In 2018, 2.4% and 20.3% of children in the general population had used hospital care and prescribed medications, respectively [7]. Apparently, children’s utilization of these healthcare services is a complex topic and relates to factors such as health conditions, social structural factors, and health insurance types. Improving child health demands a better understanding of factors linked to insured children’s utilization of physician and dentist services as well as hospital care and prescribed medications. This study is the first to examine factors in the utilization of these four healthcare services focusing exclusively on insured children in the United States.

## 2. Literature Review

In the present study, insured children’s access to healthcare services in the United States was conceptualized based on the behavioral model of health-services use [8,9,10]. The model has been frequently applied to examine determinants of healthcare utilization [11]. The model theorizes that such use is determined by perceived health need (health status), with sociodemographic characteristics (e.g., age, gender), with social structural factors (e.g., race/ethnicity, parent’s education), with financial resources (e.g., family income, health insurance), with community resources (e.g., proximity to health services), and with relevant government policies (e.g., Medicaid, State Children’s Health Insurance Program [SCHIP]) [8,9,10]. This study applied Aday and Andersen’s model to insured children’s utilization of physician visits, dentist visits, hospital care, and prescriptions in the United States.

The perceived health status of children affects their utilization of health services. In the United States, a negative association has been reported between children’s self-reported health and their likelihood of visiting a physician, using prescriptions, and using hospital care [3,12,13]; however, another study found no association between visiting a physician and children’s self-reported health [14]. Prior research on adults in the United States has found, in addition, that oral health and dental care access contributed to self-reported health, even though physical health typically tends to be considered as separate from oral health [15]. Indeed, 60% of children have rotten teeth by age 5 [16], and 6.3% to 23.3% of children’s self-reported teeth condition is considered fair or poor [17]. Moreover, a study on Latino children indicated that children’s use of dental services was promoted by parents’ belief in keeping their children’s teeth healthy [18].

In addition, children’s access to health services may be associated with their demographic characteristics. In the United States, some research has reported younger children to be more likely than older ones to visit a physician [12,13,14]; the opposite result, however, has been observed by other studies [19,20]. In addition, some research has indicated an association in the positive direction between children’s age and their likelihood of visiting a dentist [4,13,19,21], while at least one further study reported a negative age–dentist visit association [3]. Studies in the literature found younger children in the United States to be more likely than older children to use hospital care [22] and prescribed medications [13]. Published findings about gender’s role in children’s utilization of the four health services include girls’ greater tendency versus boys to use prescribed medications [13]. Moreover, some studies in the United States found girls to be more likely than boys to visit a dentist [3,4,13,19], although other studies reported no link between children’s gender and visiting a dentist [21,23]. In contrast, the literature consistently reports no association between gender and children’s likelihood of visiting a physician in the United States [13,19,23] and of using hospital care [7,13,22].

Children’s socio-structural factors are also related to their use of health services, the literature reports. Compared to White children in the United States, racial/ethnic minority children have appeared less likely to visit a physician [12,13,17,23,24] or a dentist [3,13,17,25,26] and less likely to use hospital care [13,27] or prescribed medication [13,21,28]. In addition, children whose parents have relatively more education are relatively likely to have access to healthcare services, according to the literature [29,30]. Parental education has, moreover, been found to demonstrate an association (in the positive direction) with children’s likelihood of visiting a physician [4,13] and visiting a dentist [3,4]. In contrast, one study observed no link between parental education and children’s use of hospital care or prescribed medications [13]. While parent employment increased the likelihood of children visiting a physician or dentist [6], children visiting a physician had no association with single motherhood [31].

Family financial resources also appear to be related to children’s access to health services in the United States. Compared to children from higher-income families, those from lower-income families are less likely to visit a physician [3,12,13,32] or a dentist [3,13,25] and less likely to use hospital care or prescriptions [13]. Some inconsistency characterizes the published findings on this topic, however. For example, one prior study demonstrated no link between family income and children’s likelihood of visiting a dentist [4]. Families at the upper end of the income range actually tend to be insured—as are, often, children whose families are at the lower end of that range. The latter is a reflection of two public health insurance programs—Medicaid and SCHIP—in the U.S. Medicaid offers free or low-cost healthcare services (e.g., doctor visits, hospital care, dental services, prescriptions, vision services, lab services, family planning, etc.) to families with incomes at or below 133% of the federal poverty level [33,34,35,36]; SCHIP covers low-cost healthcare services for uninsured children of families with incomes that are too high to be eligible for Medicaid but who are unable to afford private health insurance [35,37]. For both Medicaid and SCHIP, some states charge premiums and out-of-pocket costs (e.g., copayments, coinsurance, and deductibles); however, the cost for SCHIP enrollees will not exceed 5% of their family annual income [35,38].

Children covered by Medicaid and SCHIP are, studies have found, just as likely to visit a physician or dentist as children who are insured by private health insurance [3,6,12,13,39,40,41,42]. They are also just as likely to use prescriptions [5,13,28,43]. On the other hand, while some studies found that they were more likely to use hospital care [13,22,44], two other studies found the opposite result [45,46]. 

Finally, where children live in the United States is a socio-structural factor, as well as community resources involved in their healthcare. For instance, Medicaid-enrolled children living in metropolitan areas have been reported to be more likely to utilize dental care than Medicaid-enrolled children in rural areas [47]. Children’s utilization of other health services, though, has been reported by two studies to lack any urban–rural differential [13,48], while another study reported young children living in metropolitan areas to be relatively likely, versus suburban and rural children, to use health services [7].

Compared to the reviewed literature focused on the general population of children in the United States, the present study focused on insured children’s utilization of health services in the United States. It hypothesized that health status, demographic factors, social structural factors, parent characteristics, family financial resources, and type of insurance held would be linked to insured children’s use of health services. 

## 3. Method

### 3.1. Sample

The sample for the present study was extracted from the 2021 National Survey of Children’s Health (NSCH). NSCH, conducted by U.S. Census Bureau (on behalf of Health Resources and Services Administration), collected information from 50,892 children and their caregivers, regarding their health status, health insurance coverage, and access to health services in the United States; information was collected through web survey, mailed questionnaire, or telephone interviews [49]. For the present study, analyzed information was limited to that from children younger than 18 who were insured. Data from children reported to have been uninsured throughout the 12 months preceding the NSCH interview were excluded in light of such children’s very small numbers in this data set. The present study’s final sample comprised 48,660 children. This study used a public-use data set and was exempted from approval by a university institutional review board.

### 3.2. Measures

Four dichotomous outcome variables—*visited a physician*, *visited a dentist, used hospital care*, and *used prescription medication*—were employed in this study. The four represented distinct types of health-services utilization. *Visited a physician*, *visited a dentist*, and *used hospital care* quantified utilization during the 12 months preceding the interview, while *used prescription medication* quantified utilization during the 3 months preceding the interview. *Visited a physician* included both in-person visits as well as virtual visits by video or phone. The present study’s conceptualization of children’s medical need followed the behavioral model of health-services utilization. *Self-reported health* quantified the respondent health status as 5 (*excellent*), 4 (*very good*), 3 (*good*), 2 (*fair*), or 1 (*poor*) via self-reports from caregivers. Higher values suggested better health. *Teeth condition* reflected the child’s teeth as 5 (*excellent*), 4 (*very good*), 3 (*good*), 2 (*fair*), or 1 (*poor*). While *teeth condition* served as an explanatory variable for dentist visits, *self-reported health* served as an explanatory variable for the other three health services.

Several explanatory variables described sociodemographic characteristics of the sample: *girl* (versus *boy*), *child age* (in years), *single mother* (yes/no), and *parent age* (in years). Further variables characterized children’s and parents’ social structural factors reflecting the behavioral model of health-services utilization. The dummy variables *Hispanic*, *Black*, *Asian*, *other racial/ethnic minorities*, and *non-Hispanic White* denoted race/ethnicity, the latter providing the reference. *Parental education level* was measured with 9 levels: 1 (*8th grade or below*), 2 (*9th–12th grade*), 3 (*graduated high school or GED*), 4 (*vocational school*), 5 (*some college*), 6 (*associate degree*), 7 (*undergraduate degree*), 8 (*master’s degree*), 9 (*doctoral or professional degree*). *Parent employment* (yes/no) described if a parent was reportedly employed in 50 of the 52 weeks preceding interview. *Metropolitan residency* (yes/no) indicated whether the child and his/her family were residing in a metropolitan area, reflecting their proximity to health services.

Another group of explanatory variables described children’s family financial resources, specifically income and health insurance status. Three dummy variables represented the ratio of each family’s income to the federal poverty level (FPL): *family income at or below 100% of FPL*; *family income at 101–200% of FPL*; and *family income above 200% of FPL* (the reference). Three dummy variables described the health insurance status of children in the 12 months preceding the interview: *private health insurance* (the reference), *public health insurance* (enrolled in Medicaid, medical assistance, or another program of public health insurance), and *other health insurance*. 

### 3.3. Data Analysis

The present data analysis involved STATA logistic regression procedures (featuring linearized variance estimations of robust standard errors), which were selected in light of the study’s 4 dichotomous outcome variables. Preliminary analysis of tolerance statistics (≥0.60) and correlations (−0.32 ≤ *r* ≤ 0.50) suggested no multicollinearity among the explanatory variables. Final data analysis included sampling weights provided by NSCH.

## 4. Results

### 4.1. Descriptive Statistics

Of insured children in the present sample, 86.5% had visited a physician, 77.1% had visited a dentist, 2.8% had used hospital care, and 17.9% had used prescriptions (see Table 1). On average, in this study, children’s self-reported health and teeth condition measured 4.6 (“very good”) and 4.3 (“very good”), respectively. About half (48.1%) of the insured children were girls; the average age of the insured children was 8.1 years. Of the insured children, Hispanic ethnicity described 13.0%, Black 6.3%, Asian 5.6%, other racial/ethnic minorities 8.3%, and non-Hispanic White 66.8%. In other words, merely one-third of insured children were racial/ethnic minority children. Only 13.3% of parents were single mothers and the average age of parents was 41.4 years old. Parental education measured 6.2 on average (“associate degree”). About three quarters (73.4%) of the insured children and their families were residing in metropolitan areas. Of the families in the sample, 12.1% had income at or below 100% of the FPL, 15.6% had income between 101% and 200% of the FPL, and 72.3% had income above 200% of the FPL. Concerning insurance coverage, 69.2% of the insured children had private health insurance; 26.7% had public health insurance; and 4.1% had some other type of health insurance. 

### 4.2. Multivariate Analysis Results

In the present study, each of the four hypothesized models of service utilization differed significantly from the corresponding null model (Wald’s χ^2^ = 417.41–1587.91, *p* < 0.01; see Table 2). The outcome *visited a physician* was associated negatively with *self-reported health* (OR = 0.63, *p* < 0.01), *child age* (OR = 0.92, *p* < 0.01), *Hispanic* (OR = 0.80, *p* < 0.01), *Asian* (OR = 0.47, *p* < 0.01), *family income at or below 100% of FPL* (OR = 0.76, *p* < 0.01), *family income at 101–200% of FPL* (OR = 0.78, *p* < 0.01), and *other health insurance* (OR = 0.58, *p* < 0.01). In turn, the outcome was associated positively with *parental education level* (OR = 1.27, *p* < 0.01) and *metropolitan residency* (OR = 1.20, *p* < 0.01; see Column A in Table 2).

The outcome *visited a dentist* was associated negatively with *Asian* (OR = 0.64, *p* < 0.01) and *public health insurance* (OR = 0.85, *p* < 0.05), and it was associated positively with *teeth condition* (OR = 1.09, *p* < 0.01), *girl* (OR = 1.14, *p* < 0.01), *child age* (OR = 1.26, *p* < 0.01), and *parental education level* (OR = 1.11, *p* < 0.01; see Column B in Table 2).

The outcome *used hospital care* was associated negatively with *self-reported health* (OR = 0.39, *p* < 0.01), *child age* (OR = 0.95, *p* < 0.01), and *Asian* (OR = 0.40, *p* < 0.01). In turn, the outcome was associated positively with *parental education level* (OR = 1.10, *p* < 0.01) and *public health insurance* (OR = 1.93, *p* < 0.01; see Column C in Table 2).

While the outcome *used prescription medication* was associated negatively with *self-reported health* (OR = 0.36, *p* < 0.01), *Hispanic* (OR = 0.80, *p* < 0.05), *Asian* (OR = 0.46, *p* < 0.01), *family income at or below 100% of FPL* (OR = 0.74, *p* < 0.01), and *family income at 101–200% of FPL* (OR = 0.77, *p* < 0.01), such an outcome was associated positively with *child age* (OR = 1.12, *p* < 0.01), *Black* (OR = 1.35, *p* < 0.01), *parental education level* (OR = 1.09, *p* < 0.01), and *public health insurance* (OR = 1.59, *p* < 0.01; see Column D in Table 2).

## 5. Discussion

The present study found that more than 86.5% of the insured children in its sample had visited a physician—a proportion similar to that reported in two prior studies on the general population in the United States [5,6]. However, close examination of the data showed that 22.0% of insured children sought physician help through virtual visits; in other words, only 64.5% of insured children did seek such help through in-person visits. Apparently, virtual visits have been facilitating many insured children’s access to physician services during the pandemic. The present results concerning visiting a dentist, namely that 77.1% of the studied children had done so, compare well with the figure of 79.5% reported in a research study [6] that included uninsured children in its sample. Similarly, the present finding that 2.8% of insured children used hospital care and 17.9% of them had a prescription benefit hewed closely to the 2.4% and 20.3% reported in one government publication on the general population [7]. Furthermore, the majority of the insured children were White, with a family income above 200% of the FPL, and were covered by private health insurance; such findings suggest that White families with a relatively higher income level are likely to afford private health insurance for their children.

Results of the present multivariate analysis partially support the hypothesis that health status, demographic factors, social structural factors, parent characteristics, family financial resources, and type of insurance are associated with insured children’s utilization of health services. Children’s health status was associated negatively with physician visits, hospital care use, and prescription medication use, a result consistent with prior findings on the general population [3,12,13,27]; that is, it was likely that an insured child in suboptimal health would have visited a physician, used hospital care, or used a prescribed medication. On the other hand, this study found that insured children with optimal teeth condition were likely to use dental services. Such a finding tended to support the notion that parental belief in children’s healthy teeth promotes children’s use of dental services [18]. Consistently across all four tested models, Asian American children were associated negatively with outcome variables, a result consistent with prior findings on the general population [3,4,12,17,24,28]. One probable explanation of such findings is that Asian American families’ cultural beliefs often prefer traditional medical treatment and medicines to American healthcare services [50,51]. Parental education, in turn, was associated positively with all four health services, again confirming results in prior studies on physician and dentist visits in the general population [3,4,13] but contrary to the report of no such association with prescription medication use in a prior study on the general population [13]. Apparently, such findings imply the importance of parents’ education level on their children’s utilization of needed health services. 

Each of its outcome variables, the present analysis found, was linked to a unique pattern of statistically significant factors (in addition to child’s health status, teeth condition, Asian American children, and parent education level). Visiting a physician was associated negatively with child age, consistent with prior findings on the general population [12,13,14]; Hispanic children with health insurance were, in turn, associated negatively with visiting a physician, again confirming something prior studies had observed in the general population [12,13,23,24]. Furthermore, this study indicated that insured children residing in metropolitan areas were relatively likely to use physician visits, supporting the finding reported in a prior study of the general population [7]. Similarly, insured children from low-income families in the present study were not likely to use physician care, consistent with results of the general population in prior studies [3,12,13,32]. On the other hand, children insured with other health insurance were not likely to use physician visits, inconsistent with no-association findings from two prior studies on the general population [3,12]. The remaining factors showed no significant association with this outcome. Such findings imply that children who are Hispanic, relatively older, and not insured with private or public health insurance are least likely to visit a physician when they are from low-income families and residing in rural areas. The present findings also indicate that children enrolled in public health insurance are just as likely as privately insured children to visit a physician [3,6,12,39,40,41,42]. A close examination of data revealed that public health insurance’s interaction terms with family income at or below 100% of the FPL (OR = 1.53, *p* < 0.01) and with family income at 101–200% of the FPL (OR = 1.58, *p* < 0.01) had significant positive associations with the likelihood of physician visits. One plausible explanation is that the federal Affordable Care Act substantially helps low-income families to secure health insurance [52,53,54,55] and increase their utilization of health services. In fact, another close examination of data indicated that Hispanic children’s interaction term (OR = 1.64, *p* < 0.01) with family income at or below 100% of the FPL and public health insurance increased their likelihood of physician visits by 64%. 

The present study suggested that girls and older children are more likely than younger ones to have visited a dentist, as some prior studies have reported about the general population [3,4,13,19,21]. Similarly, insured children from lower-income families were as likely as those from high-income families to have visited a dentist, supporting a prior study’s results on the general population [4]. However, contrary to results of prior studies on children of the general population in a single state that showed no significant difference in the likelihood of visiting the dentist between children insured with private health insurance and those insured with public health insurance [6,42], this study found that children whose public health insurance was covered in the present study were less likely than privately insured children to have visited a dentist. Interestingly, Hispanic and Black children in the present study were as likely as insured White children to have visited a dentist, which contradicts results of prior studies on the general population [3,13,17,25,26]. Such a discrepancy may be plausibly explained by some racial/ethnic minority children’s parents of low-income families growing awareness of oral health’s consequences for children [18].

Similar to the finding of a prior study of hospital care use for nonurgent conditions on the general population [22], the present study showed that insured younger children were more likely to use hospital care than insured older children. Furthermore, the present finding indicated that children insured with public health insurance were more likely to use hospital care than those insured with private health insurance, supported by results of prior studies of the general population [13,22,44]. These findings imply that young children covered by public health insurance were likely to use hospital care. Since the study found no significant difference in children’s utilization of physician visits between having public health insurance and having private health insurance, many children with public health insurance probably used hospital care for nonurgent conditions [22,45]. Another probable explanation was that younger children with severe medical conditions may not live to be older children.

Unlike a prior study that included uninsured children [13], the present study found older children to be more likely than younger children to use prescription medication. Also contrary to prior findings on the general population [5,13,28], the present study showed that children insured with public health insurance were more likely to use prescription medication than those children insured with private health insurance. This suggests that enrollment in public health insurance does facilitate many low-income children to gain access to prescribed medication. However, the present analysis of data from low-income insured children showed their use of prescriptions to be relatively unlikely, echoing prior findings [13]. Furthermore, while present findings on the negative association between Hispanic and Asian American children and their prescription use confirm results of prior studies [13,28], the present result on the positive association between Black children and their prescription contradicts findings of prior research [13,21,28]. A closer examination of data revealed that interaction terms between family income at 101–200% of the FPL and Hispanic children (OR = 0.59, *p* < 0.05), as well as Black children (OR = 0.62, *p* < 0.05) and Asian American children (OR = 0.77, *p* < 0.01), had negative associations with the likelihood of their use of prescriptions. These findings imply that many low-income racial/ethnic minority families (especially those who were not eligible for or did not enroll in public health insurance) cannot afford prescription medication for their children. 

## 6. Conclusions

This study is the first to examine the utilization of healthcare services focusing exclusively on insured children in the U.S. Applying Aday and Andersen’s behavioral model of health-services utilization in this analysis of NSCH-2021 data revealed distinct patterns of factors associated with insured children visiting a physician, visiting a dentist, using hospital care, and using prescribed medication in the United States. Moreover, low-income children insured by public health insurance enjoyed an equal—or greater—advantage, in terms of visiting a physician, using hospital care, and using prescribed medication, compared to children insured under private plans. This means that Medicaid and SCHIP programs successfully bring healthcare access to low-income children.

From the present results, it appears that racial/ethnic disparities characterizing insured children and their utilization of health services continue; what is more, insured children of Hispanic and Asian ethnicities were, here, less likely than insured White children to use these health services in the United States. To promote racial/ethnic minority families’ access and need for health services, it is important to make racial/ethnic minority families aware of available healthcare services such as medicine discount programs underwritten by pharmaceutical manufacturers or others; some of these prescription assistance programs even offer free medication [56,57]. It is also important to understand and respect ethnic cultural assumptions and values concerning children’s health and treatment, especially among Asian American families. Moreover, similar efforts are needed to promote relatively less educated parents’ awareness of their children’s health conditions and need of access to health services.

## 7. Limitations

Two notable limitations affected this study. First, cross-sectional in nature, the present study captured participants’ health-services utilization over a very short period only. Future research might use longitudinal analysis to quantify children’s use of particular services over longer periods, which would allow for the determination of trends in or patterns of health-services use. Second, the data set employed in the present study provided no data on ethnic cultural factors, but such information would seem rather useful. Subsequent research might explore cultural or other factors perhaps underlying racial/ethnic minority children’s use of health services.

## Figures and Tables

**Table 1 ijerph-21-00427-t001:** Descriptive statistics (n = 48,660).

Variables	%	Mean	Range	sd
Outcome Variables				
Visited a physician (yes)	86.5			
(no)	13.5			
Visited a dentist (yes)	77.1			
(no)	22.9			
Used hospital care (yes)	2.8			
(no)	97.2			
Used prescription medication (yes)	17.9			
(no)	82.1			
Medical Needs				
Self-reported health		4.6	1–5	0.7
Teeth condition		4.3	1–5	0.9
Social-demographic Characteristics				
Girl	48.1			
Boy	51.9			
Child age		8.1	0–17	5.3
Single mother (yes)	13.3			
(no)	86.7			
Parent age		41.4	18–75	9.3
Social-structural Factors				
Hispanic	13.0			
Black	6.3			
Asian	5.6			
Other racial/ethnic minorities	8.3			
Non-Hispanic White	66.8			
Parental education level		6.2	1–9	1.9
Parent employment (yes)	76.3			
(no)	23.7			
Metropolitan residency (yes)	73.4			
(no)	26.6			
Financial Resources				
Family income at or below 100% of FPL	12.1			
Family income at 101–200% of FPL	15.6			
Family income above 200% of FPL	72.3			
Private health insurance	69.2			
Public health insurance	26.7			
Other health insurance	4.1			

Notes: sd = standard deviation.

**Table 2 ijerph-21-00427-t002:** Logistic regression results on insured children’s utilization of 4 healthcare services (n = 48,660).

	A		B		C		D	
Variables	Visited a Physician		Visited a Dentist		Used Hospital Care		Used Prescription Medication	
	OR	LSE	OR	LSE	OR	LSE	OR	LSE
Medical Needs								
Self-reported health	0.63 **	0.03	n/a	n/a	0.39 **	0.02	0.36 **	0.01
Teeth condition	n/a	n/a	1.09 **	0.03	n/a	n/a	n/a	n/a
Social-demographic Characteristics								
Girl	1.02	0.05	1.14 **	0.05	1.07	0.12	0.94	0.05
Boy	1		1		1			
Child age	0.92 **	0.01	1.26 **	0.01	0.95 **	0.01	1.12 **	0.01
Single mother (yes)	1.15	0.10	0.90	0.07	0.97	0.16	0.99	0.08
(no)	1		1		1			
Parent age	0.99	0.00	1.00	0.00	1.00	0.00	0.99	0.00
Social-structural Factors								
Hispanic	0.80 **	0.06	1.07	0.09	0.89	0.15	0.80 *	0.07
Black	0.86	0.09	0.85	0.08	0.89	0.19	1.35 **	0.14
Asian	0.47 **	0.04	0.64 **	0.06	0.40 **	0.09	0.46 **	0.06
Other racial/ethnic minorities	0.99	0.08	0.92	0.07	1.04	0.18	1.02	0.08
Non-Hispanic White	1		1		1			
Parental education level	1.27 **	0.02	1.11 **	0.02	1.10 **	0.04	1.09 **	0.02
Parent employment (yes)	1.09	0.07	0.99	0.05	1.17	0.14	0.93	0.06
(no)	1		1		1			
Metropolitan residency (yes)	1.20 **	0.06	0.97	0.05	1.15	0.13	1.10	0.06
(no)	1		1		1			
Financial Resources								
Family income at or below 100% of FPL	0.76 **	0.06	0.91	0.07	0.73	0.12	0.74 **	0.07
Family income at 101–200% of FPL	0.78 **	0.06	0.94	0.06	1.15	0.17	0.77 **	0.06
Family income above 200% of FPL	1		1		1			
Public health insurance	0.94	0.07	0.85 *	0.05	1.93 **	0.29	1.59 **	0.10
Other health insurance	0.58 **	0.07	0.86	0.09	1.09	0.35	1.16	0.16
Private health insurance	1		1		1		1	
Wald’s χ^2^ =	1060.73 **		866.57 **		417.41 **		1581.97 **	

Notes: ** *p* < 0.01; * *p* < 0.05; OR = odds-ratios; LSE = linearized standard errors; n/a = not applicable.

## Data Availability

This study employed solely public-use data.

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
