# Peer review of "Factors Associated with Insured Children’s Use of Physician Visits, Dentist Visits, Hospital Care, and Prescribed Medications in the United States: An Application of Behavioral Model of Health-Services Use"

_ijerph, 2024, doi:10.3390/ijerph21040427_

Round 1

Reviewer 1 Report

Comments and Suggestions for Authors

Good study and relevant findings are presented. 

A sentence on the insurance scheme- is there any premiums by the family? Were there any co-payments - will help.

As many countries across the world are developing insurance based care, consider a section on learnings for other countries.

Author Response

  1. At the beginning of Abstract, we have added the emphasis that this study is the first examined factors in utilization of physician services, dentist services, hospital care, and prescribed medications focusing exclusively on insured children in the United States. [Reviewer 2, comment #15]
  2. In the middle of Abstract, we have added “Children in the present sample were covered by private health insurance, public health insurance, or other health insurance.” Then, we have replaced "poor health to be associated..." with "self-reported health to be negatively associated with..." We also have replaced "healthy teeth to be associated with..." with "teeth condition to be positively associated with..." [Reviewer 2, comments #1 & #3]
  3. In Abstract, we have double-checked the reported directions of all associations between outcome and independent variables in the original manuscript; they are all reported in correct directions. [Reviewer 2, comment #2]
  4. Near the end of Introduction, we have added "Apparently, children’s utilization of these health care services is a complex topic and relates to factors such as health condition, social structural factors, and health insurance types." At the end of Introduction, now have added “This study is the first examined factors in utilization of physician services, dentist services, hospital care, and prescribed medications focusing exclusively on insured children in the United States”. [Reviewer 2, comments #4 & #15]
  5. We have reorganized the first paragraph (bottom of page 1) in Literature Review. We added a statement that the Aday and Andersen's model has been widely used in relevant studies, supported with a cited study. We have cited the original articles of the model to support our description of the model. We have also moved the statement "This study applied Aday and Andersen’s model to insured children’s utilization of physician visits, dentist visits, hospital care, and prescriptions in the United States” to the end of same paragraph (top of page 2). [Reviewer 2, comment #6]
  6. On page 2, second paragraph, we have rephrased “self-reported health” and self-reported teeth condition” to emphasize the measures of health and teeth conditions used by the cited studies. Moreover, have removed 2 cited studies that used illness or medical records for measure of health condition. [Reviewer 2, comments #1 & #7]
  7. On top of page 3, we now have added services covered by Medicaid and SCHIP programs, supporting with cited reports. At the end of same paragraph, we now have added that some states charge premiums and out-of-pocket costs for Medicaid and SCHIP. [Reviewer 1, comments #1 & #2]
  8. In Sample section (page 3), the second statement, we have added "NSCH, conducted by U.S. Census Bureau (on behalf of Health Resources and Services Administration)..." In other words, NSCH is not conducted by HRSA. We also have added a statement "information was collected through web survey, mailed questionnaire, or telephone interviews" in the middle of Sample paragraph”. Both statements are supported by the cited report. [Reviewer 2, comments #5 & #8]
  9. In Data Analysis section (page 4), we have emphasized that application STATA logistic regression procedures in the Data Analysis section. Also, the statistics reported in Table 2 are default statistics provided by STATA logistic regression procedures; we have replaced the mistaken label F-value with Wald's Chi-Square; p-values are indicated by asterisks which are denoted in the notes of the Table 2. [Reviewer 2, comments #10 & #11]
  10. In Discussion (page 5, middle of fifth paragraph), we did emphasize that "Apparently, virtual visits have been facilitating many insured children’s access to physician services during pandemic" in the original version. [Reviewer 2, comment #12]
  11. In Discussion (page 5, the end of fifth paragraph), we have added a statement pointing out the majority of insured children were White, from high income families, and covered with private health insurance. We also added the implication of such findings. [Reviewer 2, comment #9]
  12. On page 6, middle of third paragraph, we have further clarified that the results of cited prior studies that showed no significant difference in likelihood of visiting dentist between children insured with private insurance and those insured with public insurance. [Reviewer 2, comment #13]
  13. On page 6, at the beginning of the fourth paragraph, we have further clarified that the cited study examined hospital use for nonurgent conditions; At the top of page 7, we also have added a probable explanation of the findings as "younger children with severe medical conditions may not live to be older children." [Reviewer 2, comment #14]
  14. In Conclusion (page 7, third paragraph), we have replaced the statement "...in order to build on recent successes...all 50 states in the United States should expand ....and the related spending" with "This means that Medicaid and SCHIP programs successfully bring healthcare access to low-income children". [Reviewer 2, comment #16]
  15. In Conclusion (page 7, fourth paragraph), "For that reason, health professionals, public health workers, and social workers must maintain their efforts to highlight racial/ethnic minority families’ access and need for health services; for instances, collaborate with schools and local social service agencies in community-based initiatives" is replaced with "To promote racial/ethnic minority families’ access and need for health services, it is important..." [Reviewer 2, comment #17]
  16. In the newly added Limitations section, we now have emphasized the two limitations mentioned in the original version. [Reviewer 2, comment #18]

Reviewer 2 Report

Comments and Suggestions for Authors

See attached

Comments on the Quality of English Language

See attached

Author Response

(The authors gave the same response as above.)
